**On the role of aerosols, humidity, and vertical wind shear in the transition of shallow to deep convection at the Green Ocean Amazon 2014/5 site**

**Sudip Chakraborty[1], Kathleen A. Schiro[1], Rong Fu[1], J. David Neelin[1]**

1. Department of Atmospheric and Oceanic Sciences, University of California, Los Angeles, Los Angeles, California

**Abstract**

The preconditioning of the atmosphere for a shallow-to-deep convective transition during the dry-to-wet season transition period (August–November) is investigated using Department of Energy (DOE) Atmospheric Radiation Measurement (ARM) GoAmazon2014/5 campaign data from March 2014 to November 2015 in Manacapuru, Brazil. In comparison to conditions observed prior to shallow convection, anomalously high humidity in the free troposphere and boundary layer is observed prior to a shallow-to-deep convection transition. An entraining plume model, which captures this leading dependence on lower-tropospheric moisture, is employed to study indirect thermodynamic effects associated with vertical wind shear (VWS) and cloud condensation nuclei (CCN) concentration on pre-convective conditions. The shallow-to-deep convective transition primarily depends on humidity, especially that from the free troposphere, which tends to increase plume buoyancy. Conditions preceding deep convection are associated with high relative humidity, and low-to-moderate CCN concentration (less than the 67th percentile, 1274 $cm^{-3}$). VWS, on the other hand, shows little relation to moisture and plume buoyancy. Buoyancy estimates suggest that the latent heat release due to freezing is important to deep convective growth under all conditions analyzed, consistent with potential pathways for aerosols effects, even in presence of a strong entrainment. Shallow-only convective growth, on the other hand, shows an association with a strong (weak) low (deep) level VWS and with higher CCN concentration.

## 1.    Introduction

Deep convection is the primary source of global precipitation over the tropics and mid-latitudes (Houze, 2004) and has a large influence on extreme rainfall events like flood and droughts (Houze et al., 2015). Deep convection is also associated with strong latent heat profiles of the atmosphere (Yin et al., 2014; Schumacher et al., 2004). Investigating the meteorological parameters and suitable environmental conditions favoring the formation and evolution of deep convection is thus of interest to more accurately predict rainfall in climate models.

Climate models often exhibit large uncertainties in rainfall variability and projection over the Amazon  (Vera et al., 2006; Li et al., 2006), due in large part to the poor parameterization and an inability to simulate the formation of deep convective clouds and their evolution. Shallow and congestus convection transports moisture from the atmospheric boundary layer (BL) to the lower and middle troposphere, thus allowing for the development of deep convection (Zhuang et al., 2017; Del Genio and Wu, 2010; Jensen and Del Genio, 2006). However, many previous studies illustrate difficulties in representing the shallow-deep evolution in models (Del Genio and Wu, 2010; Waite and Khouider, 2010). Direct connections between the shallow-to-deep convection evolution and the ambient environment as well as land surface are neither fully understood nor adequately represented in climate models. There are a number of factors that can potentially dictate whether shallow convection will develop into deep, precipitating convection, such as free tropospheric moisture, vertical wind shear, cold pool formation, cloud-aerosol interactions, and the diurnal cycle.

Many studies have investigated the role of total precipitable water and moisture content
of the boundary layer (BL) on the strength and evolution of deep convections both over tropical
land and ocean sites (Schiro et al., 2016; Holloway and Neelin, 2009). In addition, there are ample
studies that show that free tropospheric moistening is important for deep convective evolution
(Waite and Khouider, 2010; Zhang and Klein, 2010; Kumar et al., 2013; Sherwood et al., 2004).
Additionally, vertical wind shear (VWS) is known to influence deep convective clouds by
influencing the slantwise ascent of the moisture (Moncrieff, 1978), separating the updraft and
downdraft regions. In a recent study, it was shown that deep tropospheric VWS (DVWS) has a
significant impact on the lifetime of mesoscale convective systems (Chakraborty et al., 2016) and
can regulate the anvil's formation  (Koren et al., 2010; Weisman and Rotunno, 2004; Petersen et
al., 2006; Kilroy et al., 2014; Harrison, 1992) as well as the updraft speed of the parcels (Weisman
and Rotunno, 2004). On the other hand, low level VWS (LVWS) can influence the rainfall and total
condensation within developing convection (Weisman and Rotunno, 2004). However, it is still
not clear how deep or lower tropospheric VWS affects updraft buoyancy. In addition, aerosols
can delay the formation of precipitation size hydrometeors, invigorating strong convection, while
suppressing shallower and weaker convection (Rosenfeld et al., 2008; Koren et al., 2008; Lin et
al., 2006; Andreae et al., 2004). Low to moderate aerosols enhance convective strength and such
an influence depends on humidity (Chakraborty et al., 2016). Furthermore, satellite data analyses
have suggested that during the dry-to-wet transition season over the Amazon, biomass burning
aerosols can increase warm clouds through their indirect effect under higher relative humidity
(RH) and moderate aerosol loading, whereas under lower RH and heavy aerosol loading
conditions, biomass burning aerosols tend to decrease clouds (e.g., Yu et al., 2006). Thus, it is
suggested that relatively small changes in the BL and in the free troposphere, due to changes of
humidity, wind profile, and aerosols can trigger or suppress deep convection. However, we lack
a clear understanding of the influence of these parameters on the deep convective evolution
from shallow convection, primarily due to observational constraints.

A few recent studies have investigated deep convective evolution and buoyancy using

ground-based measurements over the Amazonia (Zhuang et al., 2017; Schiro et al., 2016). Schiro
et al. (2016) found that given sufficient mixing in the lower troposphere, column water vapor can
be used as a proxy to understand the impact of free tropospheric humidity on plume buoyancy
related to deep convective evolution. Sensitivity of buoyancy to other factors in the Amazon was
also suggested, such as BL and microphysical processes, but the role of aerosols or VWS on deep
convective evolution from shallow clouds was not analyzed. Another study by Zhuang et al.
(2017) suggested that wind shear plays no significant role in convective evolution and that
convective available potential energy is highest during the transition period. However, they did
not assess indirect effects of vertical wind shear on the thermodynamic environment and updraft
buoyancy. Additionally, these studies primarily focus on the wet season when RH is high, yet not
explicitly on the transition season when RH is lower and aerosol concentration can be high. It is
thus unclear whether other variables, such as VWS and aerosols, influence the transition to deep
convection, either directly or by indirectly modifying the thermodynamic environment, or
whether there may be factors such as air mass source that simultaneously affect VWS or aerosols
and contributions by humidity to onset of deep convection. A key to answering these questions
might be found by analyzing the pre-convective environment. Here, we examine the association
of these variables with estimates of plume buoyancy prior to the formation of deep convection.
The DOE Atmospheric Radiation Measurement (ARM) Mobile Facility in Manacapuru,
Brazil, established as part of the Green Ocean Amazon campaign (GoAmazon2014/5) provides a
suite of ground based measurements with high spatial and temporal resolution from January
2014 to December 2015. We analyze profiles of entraining plume buoyancies and assess how
deep convection may be affected by humidity, VWS, and aerosol concentrations seasonally. Our
main interest is to assess the effects of these variables on shallow to deep  convection transition
in the dry-to-wet transition season (August-November) in an effort to shed light on factors that
control the increasing frequency of shallow to deep convection transition that drives the
monsoon onset (Wright et al., 2017).
**2.     Data and methodology**
A suite of ground based observations from the GOAmazon campaign in Manacapuru, Brazil are
employed in this study to better understand the shallow-to-deep convective transition. The main
site is located at 3°12' S, 60°35' W at 50m altitude above sea level. The data for this analysis spans
from March 2014 to November 2015. Selection of this period was based on data availability.
**2.1     Data**
The primary instrument used to distinguish between shallow and deep convection by
estimating cloud boundaries is a zenith pointing 95 GHz W-band radar, which works in both a co-
polarization and cross-polarization mode. The reflectivity data (valid range between -90 to 50
dBZ) have temporal and vertical resolutions of one second and 30 meters, respectively, that is
provided as a function of height and time in the units of dBZ with measurement accuracy of 0.5
dBZ. This dataset is available from February 2014 to November 2015. In addition to using the
radar data to identify the cloud top, we have also used the Micropulse Lidar (MPL) to co-detect
the convective tops. This is to reduce the uncertainty of the detection (as well as false detection)
of the shallow and deep clouds due to the radar attenuation problem.  The MPL is a ground-
based optical remote sensing system that determines the top and base heights of clouds using a
30 second cloud mask based on the Z. Wang et al algorithm. Based on a time-resolved signal of
transmitted and backscattered pulse, a real-time detection of the clouds can be made. These
datasets are available from January 2014 to December 2015.
Vertical profiles of thermodynamic variables, such as zonal and meridional wind speed
and direction, temperature, and relative humidity at pressure altitudes (from the surface to 3hPa)
are derived from the balloon-borne sounding system These data are available from January 2014
to November 2015 and the measurements are taken daily at 0530, 1130, 1430 (occasional), 1730,
and 2330 GMT. Radiosonde data provide information about meteorological and thermodynamic
profiles, such as humidity, temperature, wind speed and direction.
Since we are also interested in understanding the role of aerosols on the convective
transition, we have used datasets from the aerosol observing system (AOS) that provides in situ
aerosol absorption and scattering coefficients as functions of the particle size and wavelength at
the surface. The AOS also provides information about particle number concentration, size
distribution, and the chemical composition of the particles, and has a cloud condensation nuclei
(CCN) particle counter that measures the CCN concentrations at a temporal resolution of one
minute. It passes aerosol particles through thermodynamically unstable supersaturated water
vapor in a column and the water vapor condenses on the aerosol particles. Particles that grow
larger are counted. In this way, they measure the activated ambient aerosol particle number
concentration that can be activated as CCN.  We analyze CCN in this study to understand the
influence of ambient aerosols on deep convection.
**2.2    Methods**
We calculate the mean buoyancy perturbation profiles between the environment and an
entraining plume for ensembles of events in which shallow and deep convective characteristics
are defined as described below. This permits investigation of the thermodynamic effect of BL
humidity (between surface and 950 hPa), free tropospheric relative humidity (between 850 and
400 hPa), low level VWS, deep tropospheric VWS, and CCN concentrations. Low-level VWS is
defined as the difference of the mean wind speed (zonal, since meridional wind difference is
smaller) between the two 100 mb thick layers centering at 937 hPa and 737 hPa (Weisman and
Rotunno, 2004); the deep level VWS is the difference between the layers centering at the 887
hPa and 287 hPa pressure levels (Chakraborty et al., 2016; Petersen et al., 2006). We calculate
VWS by subtracting the mean wind speed of the top layer from that of the bottom layer.
We define shallow convection as having a cloud top height (CTH) below 4 km above the
surface with a convective depth of more than 2 km. Deep convection is identified when CTH
extends 8 km or more above the surface with a depth of more than 6 km (Wang and Sassen,
2007). In order to avoid errors related to the attenuated radar and Lidar pulses, we used both
the radar reflectivity (>-5 dBZ; Wang and Sassen 2007) and CTH derived from the MPL to identify
shallow and deep convection. From the radar dataset, we first separate the shallow convection
based on whether they remain shallow cloud until demise or whether they grow into deep
convection with time. Since we are interested in understanding why some shallow convection
evolves into deep convection while others do not, we investigate the meteorological,
thermodynamic, and aerosol properties before these shallow clouds form. Conditions before
shallow convection, which grows into deep convection with time, are considered to be "before
shallow-to-deep", or BSHDP. On the other hand, conditions pertaining to shallow convection
that stays shallow are considered to be "before-shallow" (BSH). For the information regarding
the profiles of RH, temperature, and wind speed during the BSH and BSHDP conditions, we use
the radiosonde measurements taken within two hours before the shallow or shallow-to-deep
convective event. CCN concentrations are averaged over ±30 minutes centered on the time of
radiosonde launch. These averaging time frames and radiosonde measurements are statistically
robust as shown in Schiro et al. (2016) where they show that temporal averaging up to and
including 3 hours yields robust statistics defining the transition to deep convection. In this study,
we show the impacts of CCNs based on 30 minutes average before and after the radiosonde
measurement. We estimate mixing ratio profiles for the BSH and BSHDP conditions from the
radiosonde data from a series of equations:
$$Vsat = 6.11 \ \times \ 10^{\frac{7.5 \times T}{237.3+T}} \qquad\qquad (1)$$

$$mrsat = \frac{621.97 \times Vsat}{P-Vsa} \qquad\qquad (2)$$

$$mr = mrsat \times RH \qquad\qquad (3)$$

where Vsat is the saturation vapor pressure, P is the pressure, T is the temperature, RH is the relative humidity, and mrsat is the saturation mixing ratio (mr) at any level.

Lastly, we evaluate the variations of entraining plume buoyancies with RH, VWS, and CCN during BSHDP and BSH events to infer the influences of these environmental conditions on the development of deep convection. The methods described in Holloway and Neelin (2009) are used here to calculate the buoyancy profiles, defined as the virtual temperature ($T_v$) differences between the environment and an entraining parcel. Buoyancies are computed using mixing and micro-physical assumptions that span a range of possibilities. Results are presented primarily for Deep-Inflow-A (DIA) mixing with and without freezing. Deep-Inflow-B" (DIB) mixing with and without freezing, and a mixing assuming constant value of the entrainment parameter are presented in the SI to test sensitivity. Parcels originate from 1000 mb and $T_v$ is interpolated in increments of 5 mb. The constant mixing case is an isobaric, fixed rate of linear mixing defined here to be 0.05 hPa$^{-1}$. DIA corresponds instead to an LES-based mixing scheme (Siebesma et al., 2007) in which the mixing coefficient depends inversely on height ($\alpha\, z^{-1}$), which has been shown to be a more realistic representation of buoyancy as compared to constant mixing (Schiro et al., 2016; Holloway and Neelin, 2009). In DIB deep-inflow mixing, mass flux increases linearly at low levels, but tapers in the mid-troposphere (Schiro et al., 2016; Holloway and Neelin, 2009). Schemes without freezing assume that the liquid water potential temperature is conserved while schemes that include freezing conserve the ice-liquid water potential temperature and all liquid is converted to ice when the plume reaches 0°C. *Schiro et al.* [2016] show results suggesting that

DIA might be a suitable scheme over the Amazon by illustrating the consistency between the
sharp increase in precipitation observed with both increasing column water vapor (CWV) and
plume buoyancies, and results are fairly similar between the two deep inflow schemes, so DIA is
presented as representative.
**3. Results**
**3.1. Mean characteristics of the BSH and BSHDP convective environments**
To identify favorable atmospheric conditions before shallow and deep convective systems
form, we evaluate differences in the mixing ratio averaged over all BSHDP (BSH) conditions
relative to such averages over all the clear sky conditions, denoted mr', in all seasons (wet, dry,
and dry-wet transition). Figure 1 shows that BSHDP conditions are associated with a higher mean
mixing ratio throughout the troposphere than BSH conditions. During the transition season, such
differences are the largest compared to the wet and dry seasons, especially above the 800 hPa
level. Differences in mr' between the BSH and BSHDP conditions can reach up to 2 g/kg at the
600 hPa level during the transition period. Additionally, mr' during BSHDP conditions is deeper
(up to 300 hPa) in the transition season as compared to the wet season (650 hPa) and dry season
(500 hPa). Differences between mr' during BSH and BSHDP conditions are smaller during the wet
season. This is likely due to the greater column moisture available throughout the wet season
(Collow et al., 2016).
Similarly, we evaluate the mean RH associated with the BSH and BSHDP conditions  at the
1000-850 hPa (lower troposphere), 850-700 hPa (lower free troposphere), 700-500 hPa (middle
troposphere), and 500-350 hPa (upper-middle troposhere) levels during all three seasons. Figure
2 shows that the pre-shallow convective conditions are associated with smaller RH compared to
BSHDP conditions for all four layers during all three seasons; however, this difference is the
strongest and most significant during the transition period above 700 hPa.
Figure 3 shows the differences in mean wind speed before the BSHDP and BSH conditions.
BSHDP conditions are associated with a change in wind speed compared to the clear sky
condition up to a height of 300 hPa, whereas BSH conditions are associated with a stronger wind
up to an altitude of 750 hPa only. This suggests that shallow convection may occur in a low level
sheared environment in comparison to clear sky conditions.
Figure 4 shows that a higher CCN concentration is associated with BSH cases in
comparison to BSHDP cases in the transition season.  It is unknown, however, whether such a
change of CCN concentration reflects aerosols' impacts on shallow to deep convection transitions
or merely an outcome of dry environments suppressing development of deep convection and or
the scavenging effect of rainfall in wet environment. The CCN levels associated with BSH are
comparable to those for clear sky or no-cloud (NC) cases, while those associated with BSHDP are
lower. For the local region of the data considered in classifying the events, the CCN observation
is prior to the convection, so local scavenging effects by wet deposition associated with
convection are excluded. However, we cannot exclude that convection-related scavenging may
have occurred upstream in the air mass prior to events, and that this could occur more frequently
under conditions that tend to lead to BSHDP events.  During the dry and wet seasons, there are
no clear and significant difference in CCN concentration between the BSHDP and BSH conditions.
**3.2.    Examining direct thermodynamic effects from humidity on buoyancy**
To examine the connection between humidity, vertical wind shear, and aerosols on the
pre-conditioning of the convective environment and how they impact the conditional instability
of the environment, we calculate buoyancies for plumes originating in the boundary layer using
simple entraining plume models. We compute differences between a plume's virtual
temperature ($T_v$) and the $T_v$ of the environment ($T_v'$) and conditionally average profiles associated
with BSH and BSHDP conditions separately based on percentiles of humidity. This allows us to
explore how the large free tropospheric moisture anomalies shown in Fig. 1 relate to the
conditional instability of the environment and prove to be favorable for the development of deep
convection, in contrast to the lower humidity observed for shallow convective cases.
Figure 5 shows that very humid free-tropospheric relative humidity (FTRH) conditions in
the upper tercile are associated with comparatively larger buoyancies during both BSH and
BSHDP conditions. Though we choose to only show results for one mixing assumption (Deep-
Inflow-A; Holloway and Neelin (2009)), this holds true under a range of mixing assumptions (as
shown in Fig. S1 of the Supplement). All BSHDP profiles are buoyant above 800 mb for any
amount of free tropospheric humidity, which highlights the success of the deep-inflow scheme
(with freezing) in capturing positive buoyancy for observed cases of deep convection. Profiles
associated with higher humidity in the upper tercile (>66.67 ‰; >70%) have significantly larger
buouyancy than other profiles. For BSH conditions (Fig. 5c), low (<33.33‰; <43%) and
moderately (33.33-66.67‰; <51%) humid environments are suitable for shallow convective
development only; however, as FTRH increases between 51% (66.67 ‰) and 71% (99.99 ‰), such
profiles appear consistent with the formation of deep convective clouds — if the plume was able
to reach to the freezing level and the release of latent heat were available for additional
buoyancy. The buoyancy profiles corresponding to instances of shallow-only convection have a
deeper layer of negative buoyancy than BSHDP cases, on average. This may be one factor acting
to suppress what may otherwise be an environment favorable for deep convection at high
humidity.
An important conclusion is that without some occurrence of freezing, the possibility of a
transition from shallow to deep convection is significantly reduced in all BSHDP cases (Betts,
1997). Here, all condensate is frozen when the parcel temperature drops below 0°C, a useful
limiting case that permits the impacts of freezing to be seen clearly. In practice, the freezing will
occur over some layer, and will depend on nucleation processes (Rosenfeld et al., 2008). Though
not explicitly tested in our analysis here, this also suggests that the effects of aerosols on freezing
microphysics are likely to be impactful to the shallow-to-deep transition. There is some sensitivity
to other entrainment schemes chosen; for instance, Deep-Inflow-B cases (Supporting Figure S1)
show positive buoyancy profiles up to 200 hPa, yet the total buoyancy is smaller compared to
that in the Deep-Inflow-A cases. These differences are attributed to the different mixing rates in
the lower free troposphere.
We also conditionally average $T_v'$ profiles by boundary layer relative humidity (BLRH) in
Figure 6. BSHDP profiles are buoyant up to 200 hPa for all BLRH values, most probably owing to
a higher RH (>72%) as compared to BSH profiles. This, again, highlights that the buoyancy
computations are successful in producing positive buoyancy for observed cases of deep
convection. As in the case of FTRH, moderate to high BLRH (>72%) is associated with larger
buoyancy for BSHDP conditions (Figure 6a), BSHDP profiles are more buoyant than BSH profiles
(Figure 6c), and consideration of freezing is a must for the deep convective evolution (Figure 6b).
On average, as seen in Figs. 1 and 2, the BL mixing ratio and BLRH (respectively) are higher for
BSHDP conditions than BSH conditions, which is also reflected in the range of values defining the
terciles in the table of Fig. 6. Though likely not the limiting factor in the transition to deep
convection, given the range of values observed for both BSH and BSHDP cases, BLRH and
buoyancy are intimately connected.
**3.3.    Examining indirect thermodynamic effects from shear and CCN on buoyancy**

Previous studies have shown that the vertical wind shear and aerosols concentration can

influence convective intensity and rainfall.  For example, VWS influences the rainfall and total
condensation within developing convection (Weisman and Rotunno, 2004), slantwise ascent of
the parcel (Moncrieff, 1978),  storm rotation, maintenance, vorticity, updraft speed (Weisman
and Rotunno, 2000), and lifetime (Chakraborty et al., 2016). Though detailed microphysical
properties are not considered in our simple plume calculations, it is worth noting that a recent
study by (Wu et al., 2017) found that lower troposheric wind shear promotes the droplet collision
and growth inside the shallow clouds by the production of turbulant kinetic energy. On the other
hand, Weisman and Rotunno (2004) using a two-dimentional vorticity simulation model found
that increasing vertical wind shear depth from surface - 3 km (low) to surface - 10 km (deep)
decreases the overall condensation and rainfall output.

However, whether and how vertical wind shear and aerosol concentrations affect the

thermodynamic environment and thus bouyancy is not well-known, especially during the
preconditioning period before the clouds form. Hence, we examine potential indirect effects of
VWS and CCN concentration on the thermodynamics of the convective environment and thus
plume bouyancy.

We look at the effect of controlling for DVWS on buoyancy profiles in Figure 7. The results

show that no significant changes in BSHDP buoyancy profiles occur through the range of DVWS
from low (3 m/s) to high (18 m/s) values (Figures 7a and 7b), which is true even for the full range
of mixing assumptions tested (not shown). However, DVWS conditions do appear related to
buoyancy among the shallow convective cases sampled. Figures 7c and 7d show that for BSH
events, buoyancy is largest in a layer between roughly 500-850 mb when DVWS is low (<33.33‰;
<3.2 m/s); as DVWS increases, buoyancy in the mid-troposphere decreases.

Recalling from Figure 3 that BSH conditions are associated with a change in wind speed

up to 750 hPa only, we also analyze the influence of the lower tropospheric VWS (LVWS). As in
the case of DVWS, controlling for changes in LVWS appears to have an insignificant influence on
the BSHDP profiles (Figures 8a and 8b). However, unlike DVWS, strong LVWS (>66.67‰; >5.64
m/s) corresponds to increased buoyancy in the lower-troposphere, especially in the 500-850
mbar layer (Figures 8c and 8d). BSH conditions associated with weak to moderate LVWS (<5.64
m/s) are associated with significantly lower buoyancy. As a result, it can be inferred that a high
LVWS or a low DVWS have associations with pre-theromodynamic conditions that might favor
shallow convection.

The role of aerosols is interesting to parse, especially because of the higher amount of

CCN concentrations associated with the BSH conditions. Figure 9 shows that low (0-33.33‰) to
moderate (33.33-66.67‰) CCN concentrations are associated with increased buoyancy above
the freezing level for the BSHDP cases than in conditions of heavy CCNs (>66.67‰, Figure 9a).
However, such an influence is not observed at altitudes below the freezing level and for BSH
conditions (Figure 9c) or when we do not consider freezing in our buoyancy computations
(Figures 9b and 9d). In Fig. 9a, the indirect effects of controlling for CCN on buoyancy above the
freezing level are notable, with the thermodynamic conditions becoming less favorable for deep
convection with increasing CCN. It is thus possible that higher CCN concentrations modify the
thermodynamic environment such that they disfavor deep convective development, even among
deep convective cases. The caveat should be noted that the results could instead imply an
association of high CCN concentrations with other factors that modify the thermodynamic
environment in this way. It is important to note that for roughly the same CCN concentrations in
the middle tercile, the buoyancy profiles for BSH and BSHDP cases are starkly different above the
freezing level. Therefore, though CCN are associated with modification of the thermodynamic
environment, an effect on the buoyancy of convective plumes, this suggests that other more
dominant variables provide leading controls on the transition to deep convection (e.g. humidity).
It is thus of interest to consider covariability between humidity and the dynamical and
microphysical variables analyzed.

In Figure 10 we calculate the conditional probability of occurrence of these conditions in

the given bin (number of samples of BSHDP and SHDP (or BSH and SH) / total number of samples
in a bin, in %) of both BSHDP and SHDP (during shallow-to-deep transitions) and BSH and SH
(during shallow convection) conditions with respect to humidity and CCN concentrations. Values
are shown only if the total number of samples in a bin is greater than 5. Figure 10a shows that
BSHDP and SHDP conditions occur predominantly above 80% FTRH. However, BSH and SH
conditions (Figs. 10 b, d, and f) occur most frequently at lower values of FTRH with a peak
probability of occurrences between 40-60% FTRH. Figure 10a shows that BSHDP and SHDP
conditions occur at high FTRH and low-to-moderate (below the 67th percentile, i.e., 0-1200
$cm^{-3}$) values of CCN concentrations. High CCN concentrations (>1200 $cm^{-3}$) (Rosenfeld et al.,
2008) and low RH (<60%) correspond to probabilities below 20%. For BSH and SH conditions
(Figure 10b), such occurrences are associated with a relatively dry (40-70% FTRH) environment
with optimal CCN concentrations ranging from 400-2000 $cm^{-3}$. This suggests that low to moderate
concentrations of CCN and high humidity are associated with deep convection.  This association
is in part qualitatively consistent with the hypothesis that high CCN concentration can reduce the
vigor of the convection by reducing the effect of convective available potential energy (*Rosenfeld*
*et al.*, 2008). Quantitatively, it should be noted that the CCN values corresponding to strong
precipitation are lower than the 1200 $cm^{-3}$ optimum for Convective Available Potential Energy
release illustrated in their buoyancy estimates. Figure 10 also has the strongest association of
BSHDP and SHDP conditions with the lowest CCN concentrations, i.e. we do not detect a
reduction at very low values with the data here. Low to moderate RH is not suitable for deep
convective buoyancy, instead favoring shallow convective development (Figs. 1-2; Fig. 10 b, d, f).
These results also suggest that CCN tend to have higher concentrations during BSH conditions.
This is potentially due to the drier environment: High aerosol concentrations owing to drier
conditions can form large numbers of small CCNs (Rosenfeld and Woodley, 2000) due to slower
coagulation and coalescence; less wet deposition would also occur due smaller probability of
precipitation. .
Consistent with the buoyancy profiles in Figs. 7 and 8, the conditional probability of
occurrence of BSHDP and SHDP also shows that VWS does not have strong impact on the
shallow to deep convective evolution (Fig. 10c, e). Again, our results suggest that higher FTRH is
a primary control in the shallow-to-deep transition.  On the other hand, shallow convection can
occur for intermediate values of FTRH (40-70%). In such conditions, low values of DVWS (<8
m/s) and appreciable LVWS (4-12m/s) are associated with conditions favorable to the
development of shallow clouds. This is consistent with increases in buoyancy observed in Figs.
7-8, though a range of conditions is depicted in Fig. 10 d and f.
We have also calculated the conditional probability of occurrences of the BSHDP as well
as SHDP as well as BSHD(SH) conditions during the wet season to provide information on
shallow-deep convective evolution during the wet season (Supporting Figure S2). In
comparison, CCN concentrations are smaller during the wet season than the transition season,
and it appears that humidity exerts the dominant control over CCN concentrations in the
evolution from shallow to deep convection (Figs. S2a and b). We do not observe any increase in
conditional probability of BSHDP events as CCN concentration increases during the wet season.
BSHDP as well as SHDP events occur at higher relative humidity during the wet season (80%-
100%, Figure S3 a, c, and e) than during the transition season (~80%, Figure 10 a, c, and e). Per
the definitions of seasons adopted here, the sample size from the dry season (May-July) is too
small to draw conclusions about the respective roles of CCN and VWS on the transition from
shallow to deep convection.
**4. Conclusion**
This study employs a suite of ground-based measurements from the DOE ARM mobile
facility in Manacapuru, BR as part of the GOAmazon campaign to investigate associations
between meteorological parameters and CCN concentrations on an entraining plume's buoyancy
before the formation of shallow or deep convective clouds during the transition season. We use
cloud radar and micropulse lidar datasets to identify shallow convection and shallow-deep
convection transitions. Radiosonde profiles measure wind speed and thermodynamic conditions
up to two hours before shallow convection develops, and the aerosol observing system measures
CCN number concentrations. Composites of CCN concentration, centered at the time of
radiosonde launch, give some indication of the association between aerosols and other
thermodynamic variables, and how these variables pre-condition the environment differently for
shallow and deep convection.
Our results show that BSHDP conditions are associated with significantly higher mixing
ratio perturbations and relative humidity above 800 hPa during the transition season compared
to clear sky conditions. Such a humid free troposphere before the development of shallow-only
clouds is not observed. Buoyancy increases as FTRH and BLRH increase for BSHDP conditions. BSH
plumes are less buoyant than BSHDP parcels owing to the fact that they occur in less humid
environments. Differences in the pre-convective humidity between the BSHDP and BSH
conditions are largest during the transition season as compared to the dry and the wet seasons.
These results suggest that moistening of the free troposphere is a necessary prerequisite for the
development of deep convection.
Excluding the buoyancy effects of freezing above the 0°C isotherm, the buoyancy is
insufficient for deep convective development, emphasizing the importance of freezing
microphysics on the shallow-to-deep convective transition. This confirms and quantifies the
potential for impacts on buoyancy by aerosol pathways operating via the freezing microphysics
(Rosenfeld et al., 2008) in presence of an important modification — the inclusion of sufficient
entrainment to give a realistic dependence on free tropospheric water vapor. Furthermore, it
confirms this potential in the range of thermodynamic environments relevant to the onset of
deep convection in the Amazon.
It is difficult to tease out a relation between dynamical and microphysical properties and
the conditional instability of the environment using plume buoyancies alone, but associations
can provide some indication of the favored environments for both shallow and deep convection.
Vertical wind shear does not appear to play a significant role in determining pre-thermodynamic
condition for the shallow to deep convective transition. However, a strong (weak) LVWS (DVWS)
appears to be related to the development of shallow convection that does not evolve to deep
convection. It is possible that this could be a causal influence of VWS, for example through the
entrainment process: if increased entrainment of dry air occurred due to a strong LVWS, it would
tend to limit the development of deep convection.  However, it could simply be a noncausal
association of conditions leading to shallow convection with those leading to strong low-level
shear. Moreover, CCN might play a different role during the transition from congestus to deep
convective evolution as shown by Sheffield et al. (2015) using the Regional Atmospheric Modeling
System. Their study shows that congestus clouds in polluted conditions are associated with
greater ice mass and strong updraft speed, unlike the shallow to deep transition cases when CCN
concentrations in upper tercile reduce the convective buoyancy. It appears that condensate
reaching the freezing level is more important for congestus – deep convective evolution than the
association of the condensate loading effect for shallow-to-deep evolution. Congestus clouds
moisten the atmosphere, reach higher altitudes than shallow clouds, and often reach beyond the
freezing level to develop into deep convection. Thus, analyses of congestus-deep convective
transition using observational data sets are needed to understand how such evolution differs
from the shallow-to-deep convective evolution discussed here.

CCNs are thought to have complex interactions with deep convection, including through

their effects on delayed rainout of small drops, latent heating associated with freezing
microphysics, and droplet evaporation. In our analysis, the probability of deep convection is
greatest in association with low-to-moderate CCN concentrations (as defined through percentiles
for the observed conditions) and high relative humidity. This is qualitatively consistent with
previous findings that suggest that aerosol microphysical effects tending to invigorate deep
convective clouds saturate and reverse as CCN concentration increases beyond ~1200/cm$^3$
(Rosenfeld et al., 2008). Corresponding effects on cloud fraction have been suggested over the
Amazon (Koren et al., 2008) for aerosol optical depth about 0.25. Higher CCN concentrations
have been proposed to slow down the autoconversion process, on the one hand potentially
permitting more condensate to reach the freezing level, but on the other adding to condensate
loading with the maximum set by competing effects on the buoyancy for deep convection
(Rosenfeld et al., 2008). The condensate loading effect of higher concentrations of CCN might
inhibit the evolution of the shallow convections into deeper convection, reducing the possibility
of deep convective transition. Our analysis shows that a higher concentration of CCN in a dry
environment is associated with BSH conditions (Figure 4).

By these mechanisms, VWS and aerosols can potentially contribute to favorable (or

unfavorable) conditions for deep convective evolutions. However, conditional instability for such
developments primarily depends on humidity and the role of aerosols and VWS warrants further
investigations. A caveat quantified here that does not seem to have been taken into account in
other studies is that data stratified by conditions on aerosol or VWS concentrations can have
substantial relationships with buoyancy that arise entirely from the thermodynamic
environment. When making inferences about aerosol impacts using techniques that seek
relationships between cloud or precipitation properties, we recommend controlling for or at
minimum quantifying such covariability.

This study advances our capability to understand how some shallow convection evolves

to deep convection and under what meteorological parameters and CCN concentrations such
evolutions are favorable during the transition season over the Amazon. High FTRH and BLRH are
required for a shallow-deep convective evolution during the transition season, which is
associated with low-moderate concentrations of CCN. Deep convection appears unrelated to
vertical wind shear in the transition season, whereas shallow convection has a weak association
to strong LVWS and weak DVWS. It is worth nothing that the results of this study may differ across
different regions. Use of different ACRIDICON-CHUVA datasets to test consistency with the
southern Amazon, which is more prone to drought conditions, could prove to be a useful
comparison.

**ACKNOWLEDGEMENTS.** This research was supported by the Office of Biological & Environmental Research within the Department of Energy (DOE), Office of Science, (DE-SC0011117 & DE-SC0011074). JDN and KAS were also supported under National Science Foundation Grant AGS-1505198 and National Oceanic and Atmospheric Administration Grant NA14OAR4310274. We acknowledge the providers of the DOE ARM datasets.

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

**Figure Legends**

**Figure 1.** Differences in the mixing ratio (mr') averaged over the before shallow (BSH) and before shallow-deep (BSHDP) conditions relative to that averaged over clear sky conditions during the a) wet, b) dry, and c) transition periods. Error bars show two-standard deviations of the data.

**Figure 2.** Mean RH of different levels during the BSH and BSHDP conditions. Error bars show two-standard errors of the data.

**Figure 3.** Differences in wind speed prior to BSHDP and BSH conditions during the transition period compared to the clear-sky condition. Solid lines represent the mean and the dotted lines represent the two-standard deviations of the wind speed for BSHDP and BSH cases.

**Figure 4.** Mean CCN concentrations (cm$^{-3}$) for the BSH, BSHDP, and clear-sky (NC) conditions over 30 minutes during all three seasons. Error bars show two-standard deviations of the data.

**Figure 5.** Profiles of delta Tv for BSH and BSHDP conditions under different cases of mixing and entrainment schemes compared to the mean environmental Tv condition obtained from the radiosonde data for different percentiles of free tropospheric RH (850-400 hPa) associated with the convections during the transition seasons. Shaded area represents two - standard errors for each profile. Values of corresponding FTRH are shown in the table. Total number of samples of BSHDP and BSH cases are 37 and 29, respectively. Solid (light blue shade), dotted (moderate blue shade), and dashed (dark blue shade) lines represent the conditionally averaged delta Tv values (two sigma error intervals) for the 0‰-33.33‰, 33.33‰-66.67‰, and 66.67‰-99.99‰ intervals, respectively.

**Figure 6.** Same as in Figure 5, but for different percentile values of BLRH. Values of corresponding BLRH are shown in the table.

**Figure 7.** Same as in Figure 5, but for different percentile values of deep tropospheric VWS. Values of corresponding DVWS are shown in the table.

**Figure 8.** Same as in Figure 5, but for different percentile values of lower tropospheric VWS. Values of corresponding LVWS are shown in the table.

**Figure 9.** Same as in Figure 5, but for different percentile values of CCN concentration. Values of corresponding CCN concentrations are shown in the table.

**Figure 10.** Contours of conditional probability (%) of (a, c, and e) BSHDP as well as SHDP; and (b, d, f) BSH as well as SH conditions with respect to (a), (b) FTRH and CCN concentrations, (c),(d) FTRH and DVWS, and (e),(f) FTRH and LVWS. Conditional probability of these conditions occurring in a given bin are estimated by dividing the number of samples of BSDHP and SHDP (or BSH and SH) conditions by the total number of samples in that bin. Blank areas correspond to bins for which neither shallow-deep nor shallow clouds are observed or total number of samples in that bin is less than 5. Total number of samples of BSHDP we well as SHDP, BSH as well as SH, and all the conditions (including clear sky) are 71, 49, and 565, respectively.

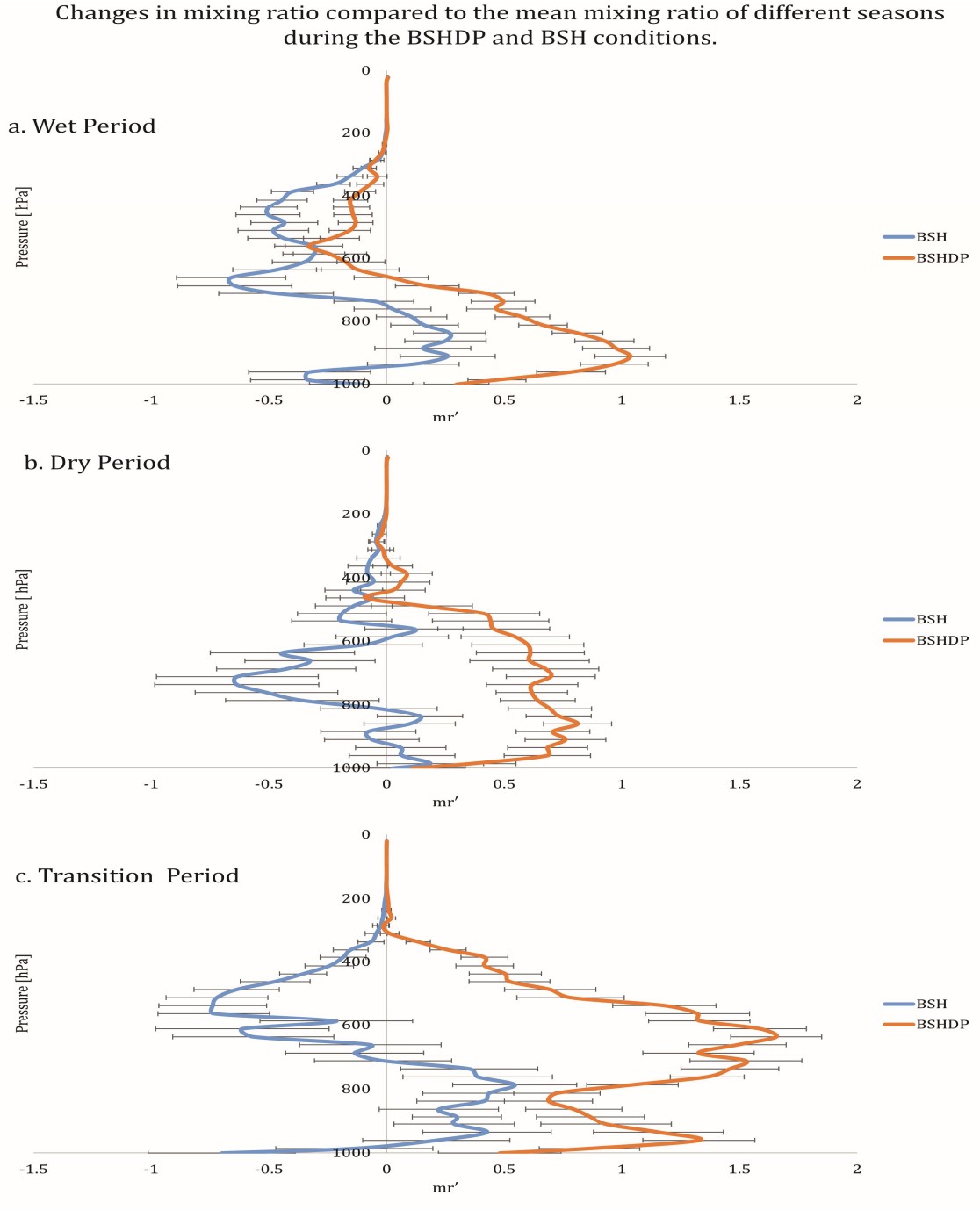

Changes in mixing ratio compared to the mean mixing ratio of different seasons during the BSHDP and BSH conditions.

**Figure 1**

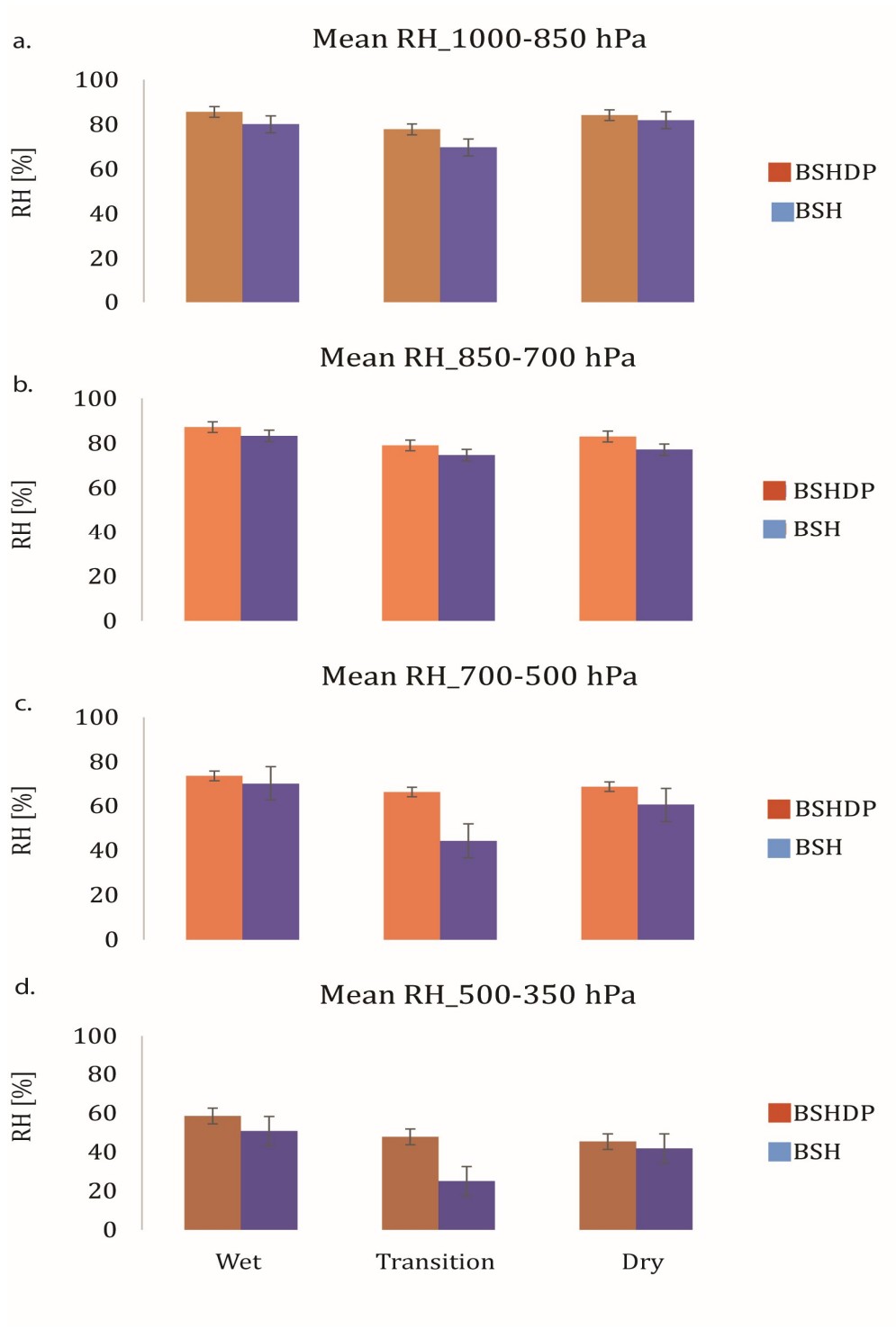

**Figure 2**

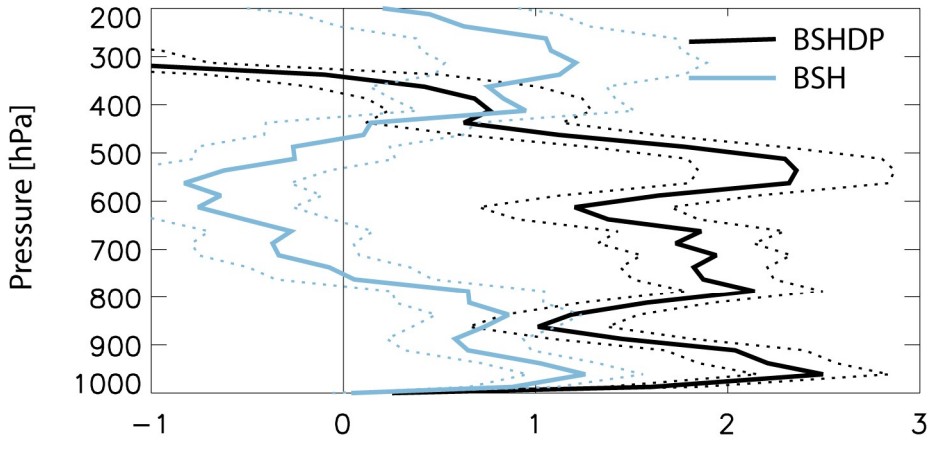

**Figure 3**

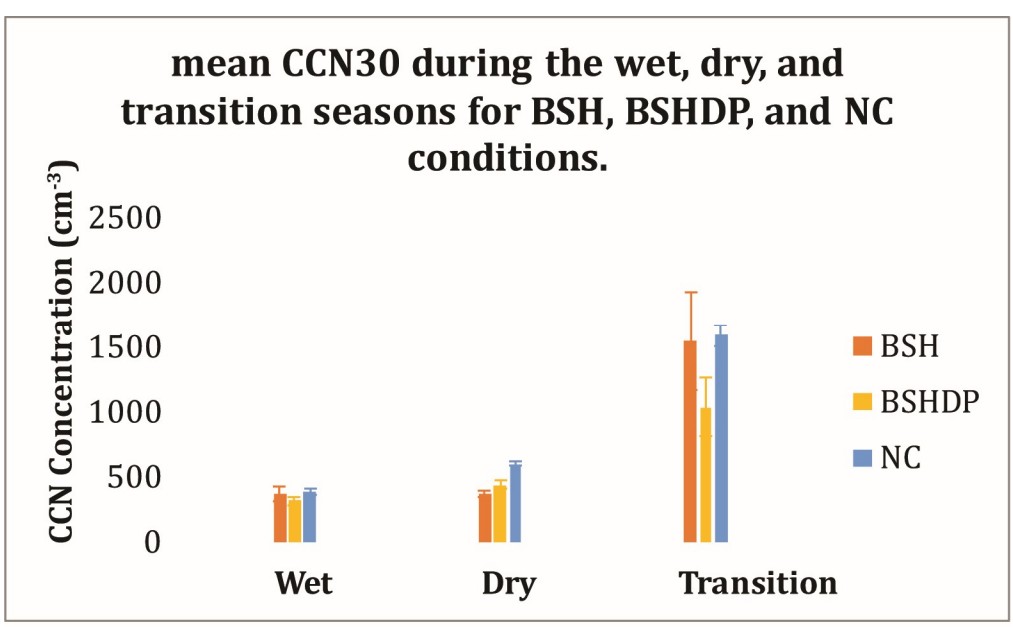

**Figure 4**

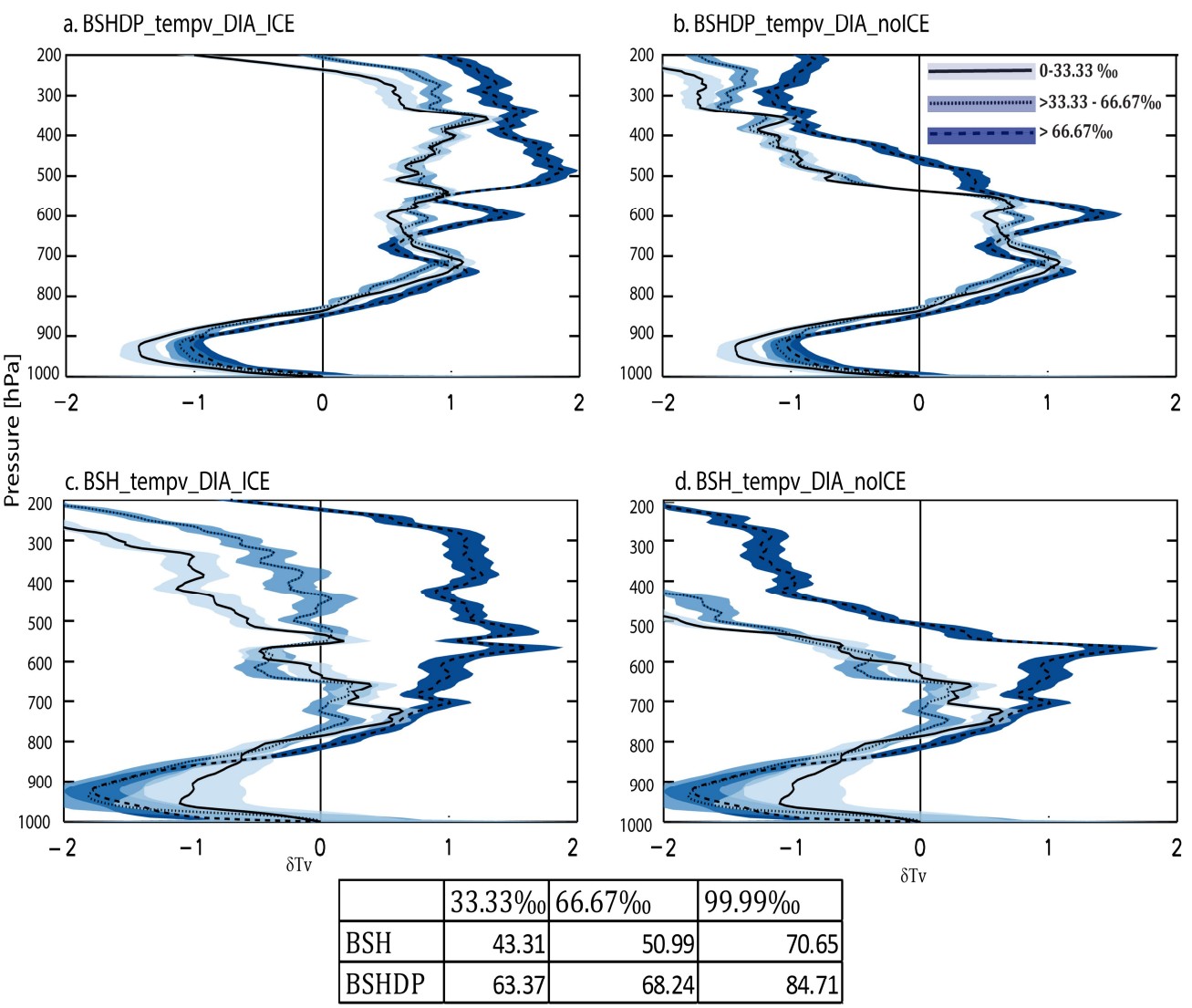

| | 33.33‰ | 66.67‰ | 99.99‰ |
|-------|--------|--------|--------|
| BSH | 43.31 | 50.99 | 70.65 |
| BSHDP | 63.37 | 68.24 | 84.71 |

**Figure 5**

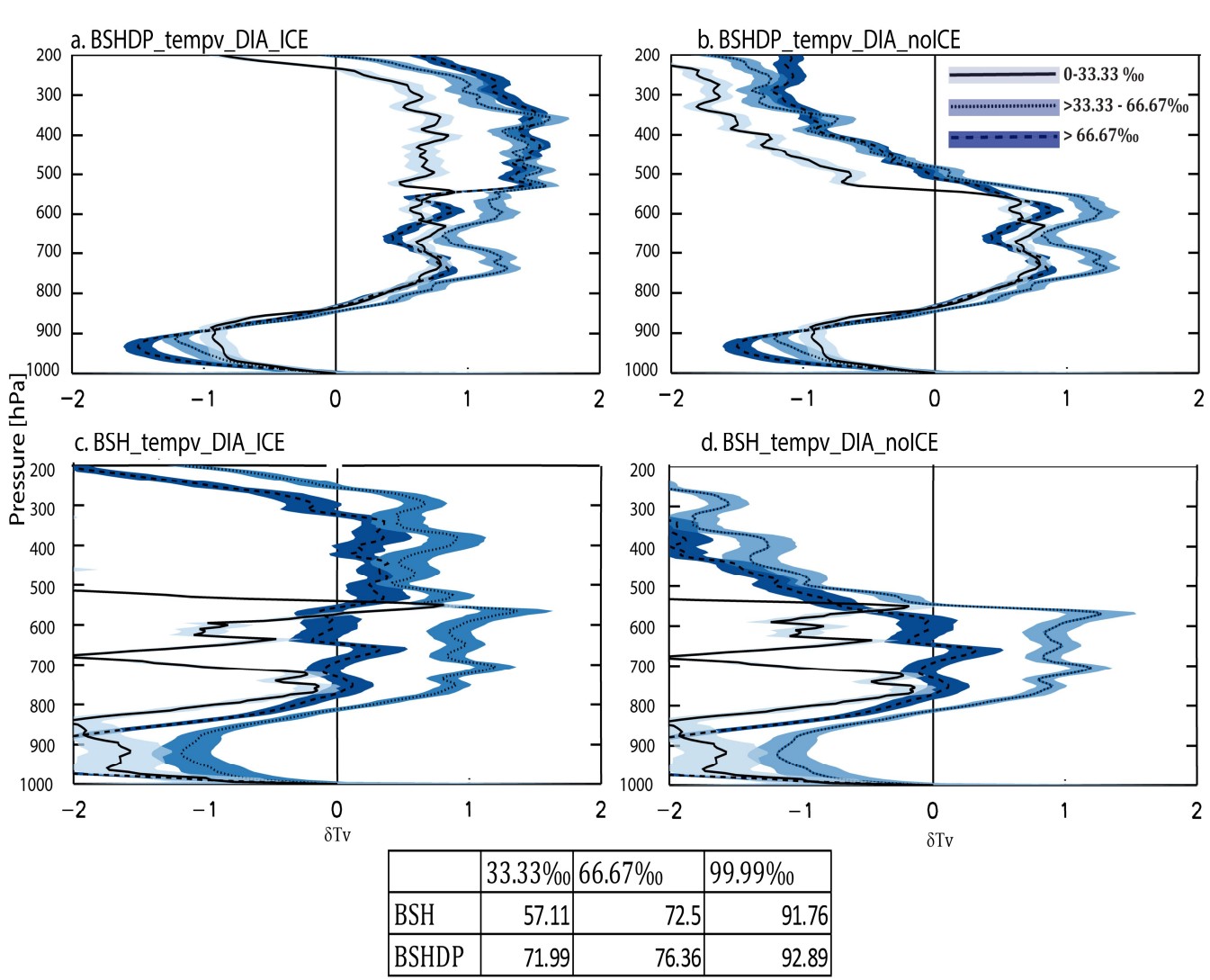

| | 33.33‰ | 66.67‰ | 99.99‰ |
|---|---|---|---|
| BSH | 57.11 | 72.5 | 91.76 |
| BSHDP | 71.99 | 76.36 | 92.89 |

**Figure 6**

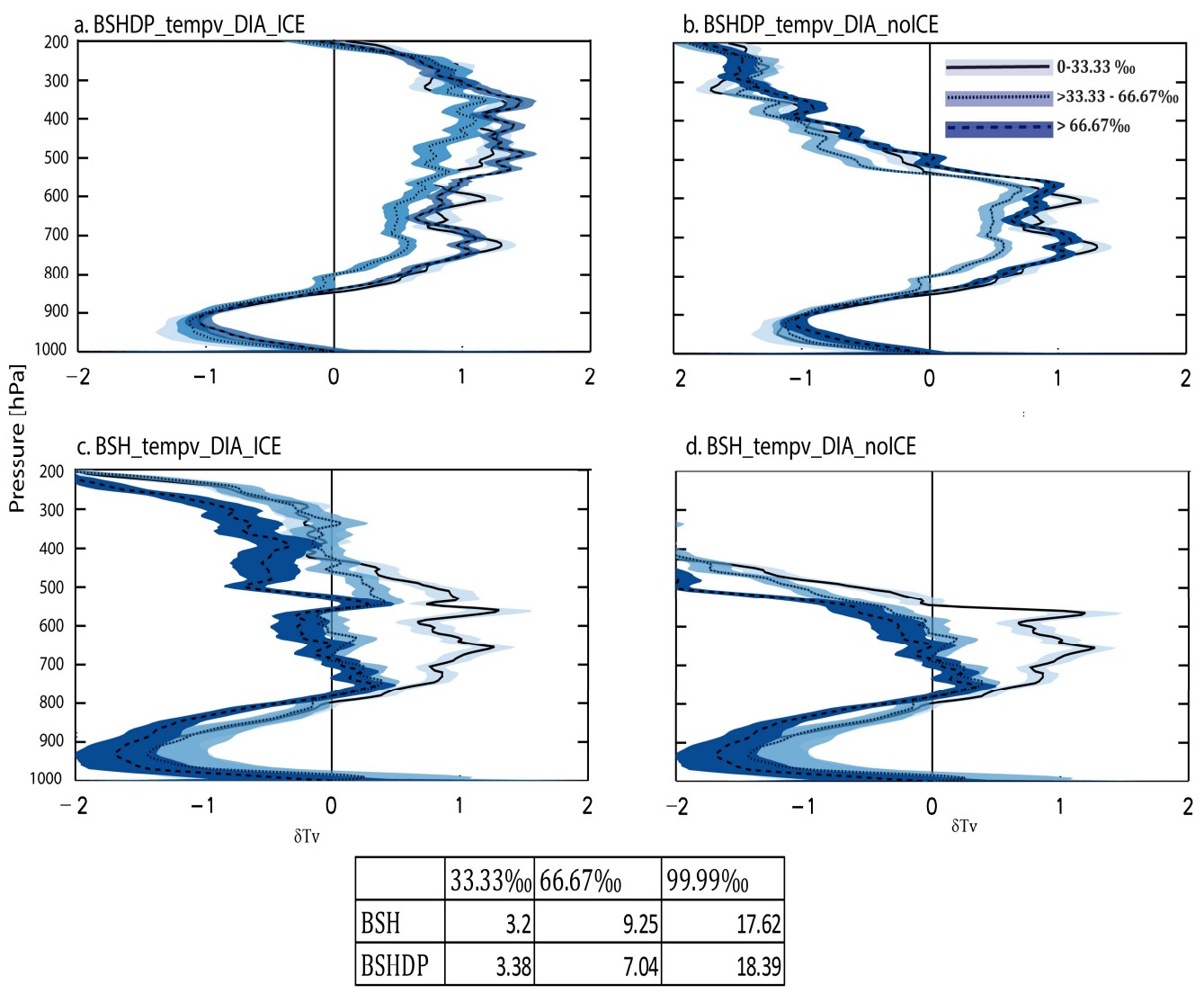

| | 33.33‰ | 66.67‰ | 99.99‰ |
|---|---|---|---|
| BSH | 3.2 | 9.25 | 17.62 |
| BSHDP | 3.38 | 7.04 | 18.39 |

**Figure 7**

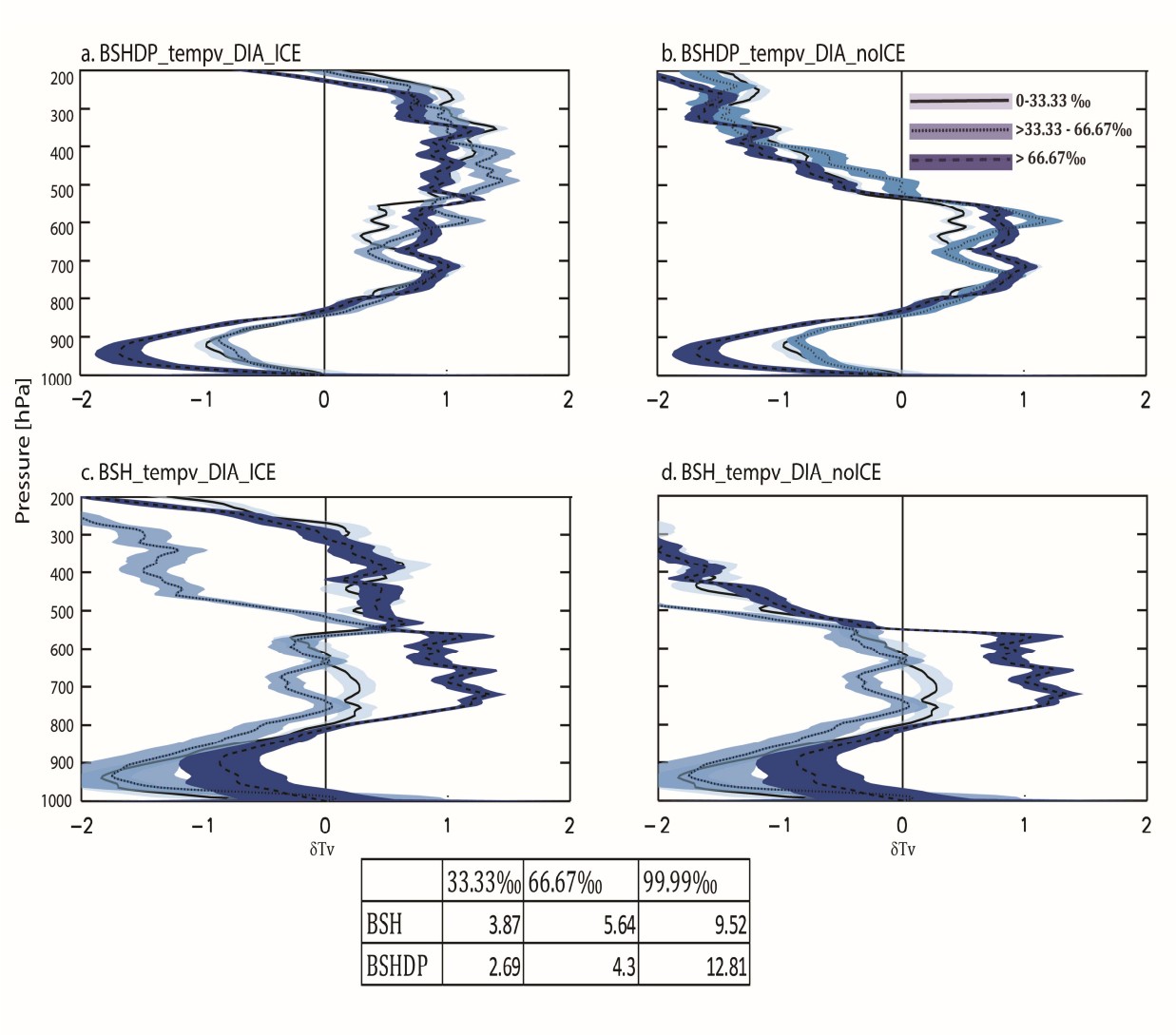

| | 33.33‰ | 66.67‰ | 99.99‰ |
|---|---|---|---|
| BSH | 3.87 | 5.64 | 9.52 |
| BSHDP | 2.69 | 4.3 | 12.81 |

**Figure 8**

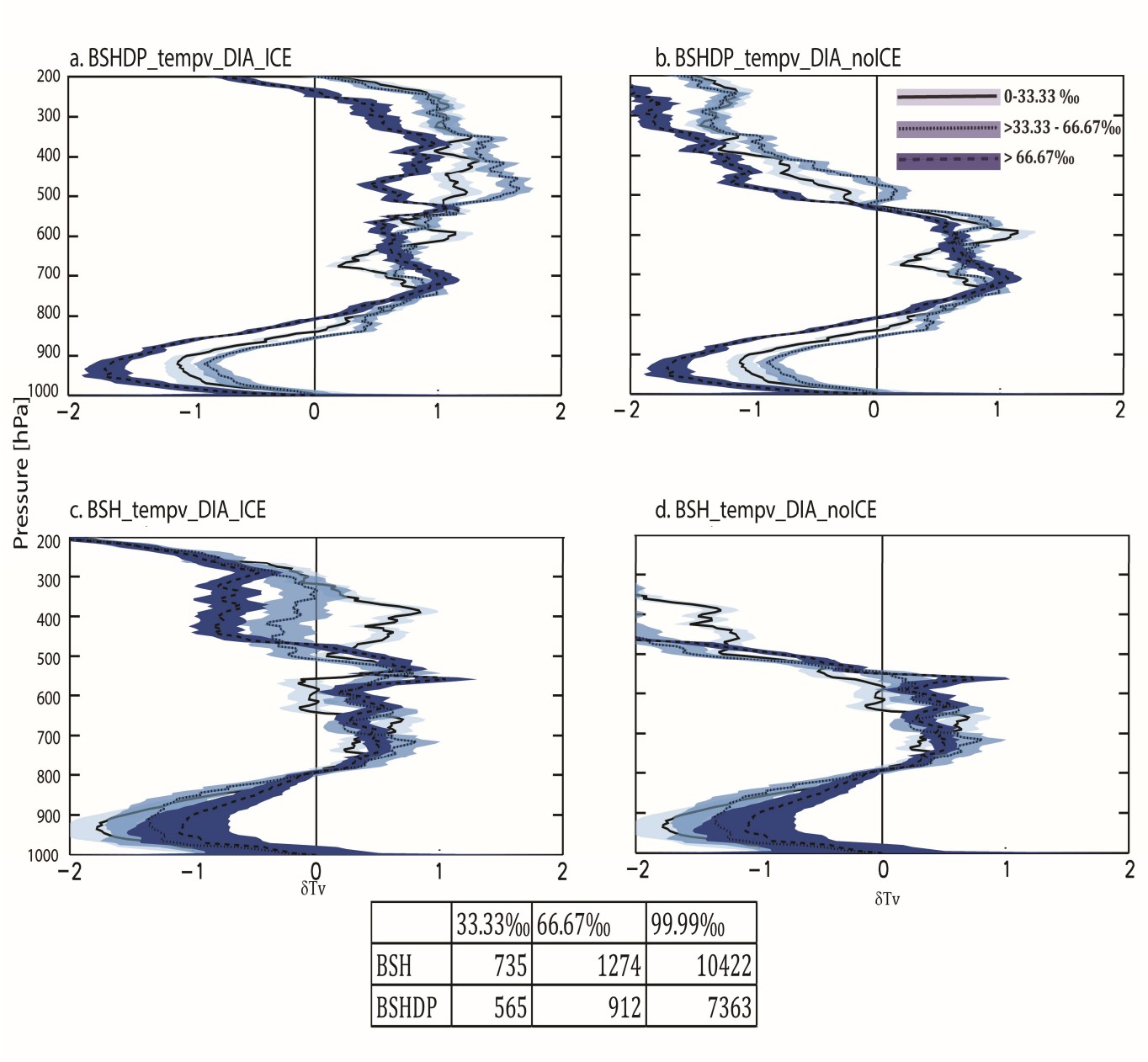

**Figure 9**

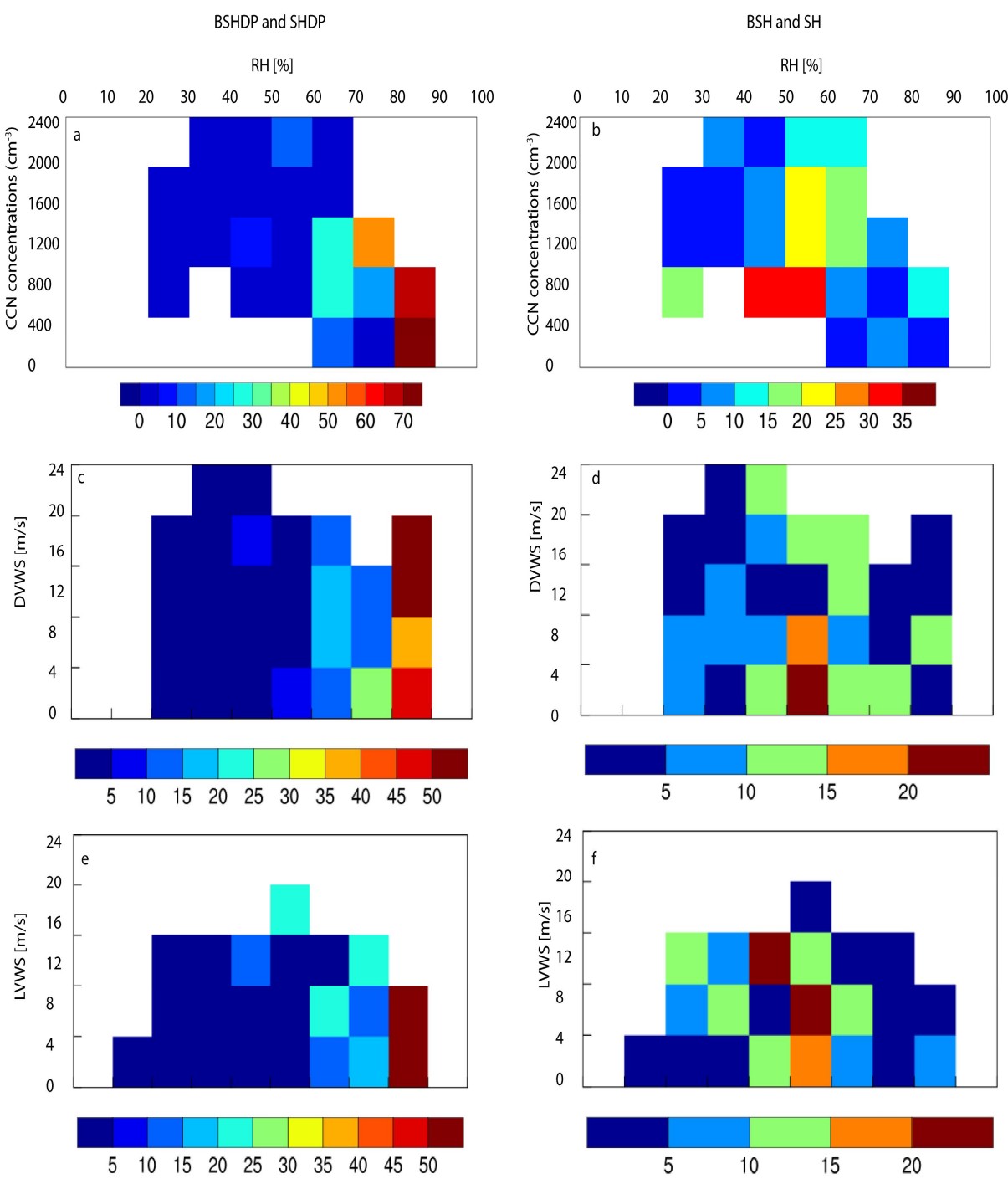

**Figure 10**