# Peer review of "Sudip Chakraborty1, Kathleen A. Schiro1, Rong Fu1, J. David Neelin1"

_Atmospheric Chemistry and Physics, 2018_

## Referee Comment (RC1) · Anonymous Referee #1 · 20 Apr 2018

Manuscript Number: ACP-2018-249

Title: On the role of aerosols, humidity, and vertical wind shear in the transition of shallow to
      deep convection at the Green Ocean Amazon 2014/5 site

Author(s): Sudip Chakraborty, Kathleen A. Schiro, Rong Fu, J. David Neelin

**General Comments:**

This manuscript reports on a study of using the GOAmazon data, together with an entraining plume model, to diagnose the role of humidity, vertical wind shear and aerosols in the transition of shallow to deep convection at the Green Ocean Amazon 2014/5 site. The results from the study show that the shallow to deep convective transition observed at the site primarily depends on humidity in the troposphere, which tends to increase plume buoyancy. Conditions preceding deep convection are associated with high relative humidity, and low-to-moderate CCN concentration. Vertical wind shear is shown to have little relation to moisture and plume buoyancy, while the latent heat release due to freezing is shown to be important to deep convective growth under all conditions analyzed. Shallow convection growth, on the other hand, shows an association with a strong (weak) low (deep) level vertical wind shear and with higher CCN concentration. The presentation of the manuscript is concise and clear, but I found that the results do not include new scientific findings. However, since the study demonstrated a useful example of the GOAmazon data analysis, I recommend that the manuscript be accepted for publication after it is revised to address the following few comments of mine.

**Specific Comments:**

Multiple places thought the manuscript: A space after semi column is needed to separate the references (e.g., in line36).

Line 69: "A few recent studies" instead of "Few recent studies"

Lines 166-168: Replace "x" with multiplication symbol "×".

Line 192: Remove letter "T".

Section 3.2: Is there any difference in the direct thermodynamical effects from humidity and buoyancy between the wet, dry and transition seasons? There is a need to separate the analysis between the seasons.

Line 288: Remove the extra tab/indentation.

Lines 247-248: Consider replacing "Profile associated with stronger humidity" with "Profile associated with higher humidity in the upper tercile".

Line 248: Replace "stronger humidity" with "higher humidity".

Lines 253-256: You should calculate the values of convective inhibition for these buoyancy profiles to support the statement.

Lines 257-258: Is this a new finding? Please cite the previous publications in this regard to compare this finding about the importance of freezing in the development of convection.

Lines 296-306: Citations of previous observational and modeling studies on the dynamical connection between the vertical wind shear and the intensity of convection should be included in the discussion.

Lines 307-325: The transition of shallow to deep convection takes places in all the wet, dry and transition seasons. Yet, CCN concentrations are sharply different between the transition and dry/wet seasons. There is a need to separately show the results from the buoyancy and covariability analyses for the three seasons to disentangle the complexity in the interaction between the aerosol loading and convection invigoration. In particular, the results with respect to the shallow convection in all the seasons should be compared with those presented in the following paper:

Sheffield, A. M., S. M. Saleeby, and S. C. van den Heever (2015), Aerosol-induced mechanisms for cumulus congestus growth, J. Geophys. Res. Atmos., 120, 8941–8952, doi:10.1002/2015JD023743.

---

## Referee Comment (RC2) · Anonymous Referee #2 · 24 Apr 2018

In this paper the transition between shallow to deep convection during the wet-to-dry season is examined. They show elevation in the humidity levels prior to the development of deep convection. When examining the links to winds and CCN concentrations they show sensitivity of mostly the shallow convection.

This study can add a measurement-base reference for the convection cycle over the tropical rain-forest. As such more details should be provided about the frequency of measurements. Information about how many blooms where used altogether? When averaging, how many profiles are used? How large the variance is? What exactly meteorological ground-based data is used and does it provide information above the

surface (profiles)? How well the surface CCN measurements reflects the conditions near cloud base?

This study focuses on the dry-to-wet season, while (to the best of my knowledge) the ARM measurements covered all seasons. Since, in my opinion, the strength of this paper is on the direct, detailed measurement approach, why not providing information on transitions during other seasons using the same methodology?

Technical comments P8 L140: Add space in "hPa(Weisman and Rotunno, 2004);". P10 L168 and L170: You denote mixing ratio as "MR". In the figures you denote it as "mr". Please correct and be consistent. P11 L191: You use the acronym "CWV" without first using the full term. P11 L191: What does "T" stand for at the end of the line? If it's a mistake, please correct it. P11 L209: You define 500-300 hPa as the upper-middle troposphere. In figure 2 panel d. it's defined differently (500-350), please correct it. P17 L335-336: The units slipped to the next line. In the figures you use different colors for the BSHDP and BSH conditions (blue, red, black, and orange). For example, you use red color for BSHDP and the blue for BSH in figure 1, but the other way around as in figure 2. I've found it confusing to the reader, please be consistent. Figures 1-4: What do the error bars stand for? Figure 3: Please add an explanation clarifying what the dashed and solid curves stand for. Figure 4: According to the text (P12 L219) the y-axis is CCN concentration, in the figure itself you just write "CCN". Please add the "concentration" and the units to the y-axis and to the figure caption. Figure 5-9: Please use bigger font size. Also, if you use colors, please explain in the legend or in the figure caption what do they stand for. Figure 10: Same note as for figure 4 for panels a and b. How many cases were included in each bin?

---

## Author Comment (AC1) · 22 Jun 2018

*We thank reviewer for their constructive comments to improve the manuscript. Our point-by-point response is given after the comments of the reviewer.*
* * *
**Reply to Reviewer #1:**

General Comments:
This manuscript reports on a study of using the GOAmazon data, together with an entraining plume model, to diagnose the role of humidity, vertical wind shear and aerosols in the transition of shallow to deep convection at the Green Ocean Amazon 2014/5 site. The results from the study show that the shallow to deep convective transition observed at the site primarily depends on humidity in the troposphere, which tends to increase plume buoyancy. Conditions preceding deep convection are associated with high relative humidity, and low-to-moderate CCN concentration. Vertical wind shear is shown to have little relation to moisture and plume buoyancy, while the latent heat release due to freezing is shown to be important to deep convective growth under all conditions analyzed. Shallow convection growth, on the other hand, shows an association with a strong (weak) low (deep) level vertical wind shear and with higher CCN concentration. The presentation of the manuscript is concise and clear, but I found that the results do not include new scientific findings. However, since the study demonstrated a useful example of the GOAmazon data analysis, I recommend that the manuscript be accepted for publication after it is revised to address the following few comments of mine.

*We thank the reviewer for very helpful comments.*

Specific Comments:
Multiple places thought the manuscript: A space after semi column is needed to separate the references (e.g., in line36).

*We have added a space after semi colon throughout the manuscript.*

Line 69: "A few recent studies" instead of "Few recent studies"

*Added, line 68.*

Lines 166-168: Replace "x" with multiplication symbol "×".

*Replaced with "×" symbol. Lines 164-166*

Line 192: Remove letter "T".

*Removed, line 190*

Section 3.2: Is there any difference in the direct thermodynamical effects from humidity and buoyancy between the wet, dry and transition seasons? There is a need to separate the analysis between the seasons.

*We have added the conditional probability of the BSHDP and SHDP, as well as the BSH and SH conditions in Figure S2 of the supplement (discussion in lines 365-376) for the wet season. Per our definitions of wet season and transition season here, there are not sufficient remaining samples for the dry season (May-July).*

Line 288: Remove the extra tab/indentation.

*Removed, line 295.*

Lines 247-248: Consider replacing "Profile associated with stronger humidity" with "Profile associated with higher humidity in the upper tercile".

*Replaced. Lines 244-245.*

Line 248: Replace "stronger humidity" with "higher humidity".

*Replaced. Line 245.*

Lines 253-256: You should calculate the values of convective inhibition for these buoyancy profiles to support the statement.

*We've removed the original discussion of convective inhibition and made a qualitative statement about the existence of negative buoyancy. Lines 251-252.*

Lines 257-258: Is this a new finding? Please cite the previous publications in this regard to compare this finding about the importance of freezing in the development of convection.

*Reference (Betts, 1997) added. Line 256.*

Betts, A. K.: The parameterization of deep convection., Nato Adv Sci I C-Mat, 505, 255-279, 1997.

Lines 296-306: Citations of previous observational and modeling studies on the dynamical connection between the vertical wind shear and the intensity of convection should be included in the discussion.

*VWS influences the rainfall and total condensation within developing convection (Weisman and Rotunno, 2004), slantwise ascent of the parcel (Moncrieff, 1978), storm rotation, maintenance, vorticity, updraft speed (Weisman and Rotunno, 2000), and lifetime (Chakraborty et al., 2016). Though detailed microphysical properties are not considered in our simple plume calculations, it*

is worth noting that a recent study by (Wu et al., 2017) found that lower troposheric wind shear promotes the droplet collision and growth inside the shallow clouds by the production of turbulant kinetic energy. On the other hand, Weisman and Rotunno (2004) using a two-dimentional vorticity simulation model found that increasing vertical wind shear depth from surface - 3 km (low) to surface - 10 km (deep) decreases the overall condensation and rainfall output. Discussion added. Lines 280-289.

Lines 307-325: The transition of shallow to deep convection takes places in all the wet, dry and transition seasons. Yet, CCN concentrations are sharply different between the transition and dry/wet seasons. There is a need to separately show the results from the buoyancy and covariability analyses for the three seasons to disentangle the complexity in the interaction between the aerosol loading and convection invigoration. In particular, the results with respect to the shallow convection in all the seasons should be compared with those presented in the following paper:
Sheffield, A. M., S. M. Saleeby, and S. C. van den Heever (2015), Aerosol-induced mechanisms for cumulus congestus growth, J. Geophys. Res. Atmos., 120, 8941–8952, doi:10.1002/2015JD023743.

*We have provided the conditional probability analysis using wet season samples in Figure S2 (lines 226-227 and 365-376). We have also added the necessity of analyzing such evolutions for congestus-deep convection cases in lines 416-426.*
* * *
**Reply to Reviewer #2:**

In this paper the transition between shallow to deep convection during the wet-to-dry season is examined. They show elevation in the humidity levels prior to the development of deep convection. When examining the links to winds and CCN concentrations they show sensitivity of mostly the shallow convection.

This study can add a measurement-base reference for the convection cycle over the tropical rain-forest. As such more details should be provided about the frequency of measurements. Information about how many blooms where used altogether? When averaging, how many profiles are used? How large the variance is? What exactly meteorological ground-based data is used and does it provide information above the surface (profiles)? How well the surface CCN measurements reflects the conditions near cloud base?

*First, we would like to thank the reviewer for very helpful comments. We have provided the number of samples using the Figure caption of Figure 5. Instead of variances, we have also provided the standard errors associated with each profile in shades. We have added information about the ground based data and the information above the surface in lines 115 and 118-119. Aerosols data is obtained at the surface (lines 120-123). Since this study focuses on*

*preconditions (up to two hours before the clouds are formed, line 149-162), we are unable to provide the information on how CCN concentrations reflect the conditions near cloud base.*

This study focuses on the dry-to-wet season, while (to the best of my knowledge) the ARM measurements covered all seasons. Since, in my opinion, the strength of this paper is on the direct, detailed measurement approach, why not providing information on transitions during other seasons using the same methodology?

*We have added the conditional probability of the BSHDP and SHDP, as well as the BSH and SH conditions in Figure S2 of the supplement (discussion in lines 365-376) for the wet season. Per our definitions of wet season and transition season here, there are not sufficient samples of BSHDP for the dry season (May-July).*

Technical comments

P8 L140: Add space in "hPa(Weisman and Rotunno, 2004);".

*Space added. Line 138*

P10 L168 and L170: You denote mixing ratio as "MR". In the figures you denote it as "mr". Please correct and be consistent.

*Changed throughout the manuscript.*

P11 L191: You use the acronym "CWV" without first using the full term.

*Thanks for pointing this out. We have used the full term. Line 188.*

P11 L191: What does "T" stand for at the end of the line? If it's a mistake, please correct it.

*Removed, line 190*

P11 L209: You define 500-300 hPa as the upper-middle troposphere. In figure 2 panel d. it's defined differently (500-350), please correct it.

Corrected. Line 207.

P17 L335-336: The units slipped to the next line. In the figures you use different colors for the BSHDP and BSH conditions (blue, red, black, and orange). For example, you use red color for BSHDP and the blue for BSH in figure 1, but the other way around as in figure 2. I've found it confusing to the reader, please be consistent.

*Units in line 339. Color changed to make figure 1 and 2 consistent.*

Figures 1-4: What do the error bars stand for?

*Mentioned in all the Figure legends that they represent the standard deviations or errors.*

Figure 3: Please add an explanation clarifying what the dashed and solid curves stand for.

*Mentioned in the edited Figure caption that they denote the mean and two standard deviation of the wind speed.*

Figure 4: According to the text (P12 L219) the y-axis is CCN concentration, in the figure itself you just write "CCN". Please add the "concentration" and the units to the y-axis and to the figure caption.

*We have added concentration after CCN in the figure.*

Figure 5-9: Please use bigger font size. Also, if you use colors, please explain in the legend or in the figure caption what do they stand for.

*We have made the fonts little bigger and also added the colors used to show two standard errors in the Figures.*

Figure 10: Same note as for figure 4 for panels a and b. How many cases were included in each bin?

*We have added the word concentrations and mentioned the number cases used in Figure 10 and S2.*